# Developing a pricing model for general medical consultation services among private consulting rooms in Harare, Zimbabwe

Chengetedzai Gota[ORCID]*, Gibson Mandozana, Richard Makurumidze, Shepherd Shamu

Department of Global, Public Health and Family Medicine, University of Zimbabwe, Harare, Zimbabwe

* chengegota@gmail.com

## Abstract

### Background

There are often variations around setting tariffs or prices for payment of general medical consultation services between the medical insurance industry, the medical association, and the private health care providers. Differences in these tariffs have an impact on both the public and private health care sectors. The research study aimed to develop a pricing model for consultation services and ascertain the ideal price that should be charged for those services.

### Materials and methods

An analytical cross-sectional study design was utilized to collect data through a questionnaire for the input costs of operating a surgery as well as the consultation fee charged by General Medical practitioners who participated in the study. The recruitment period for this study was from 13 September 2023, to 29 September 2023, during which 170 medical practitioners completed the online questionnaire. Eight independent variables were analyzed in the study using Stata (version 14) to build a multiple linear regression model with the capability of predicting the mean consultation fee. The variables were registration fees, rental, cost of equipment, cost of consumables, salaries, utility costs, number of patients seen and actual profit.

### Results

A total of 170 General Medical Practitioners participated in the study. The variables which met the multiple linear regression assumptions that were included in the model were consumables, salaries, utilities, number of patients seen by doctor and actual profit made by the consulting room. The estimated ideal consultation fee, which was obtained using empirical data from the 170 surgeries sampled was US$23.28. Profit levels varied significantly by both suburb density ($p < 0.001$) and patient volume

**Data availability statement:** All relevant data are within the manuscript and its Supporting Information files.

**Funding:** The author(s) received no specific funding for this work.

**Competing interests:** The authors have declared that no competing interests exist.

($p < 0.001$), indicating the importance of geographic and operational factors in practice profitability.

## Conclusion

The developed model aimed to provide a transparent basis for determining consultation fees, thus helping to assure affordability, access, and financial sustainability for both practitioners and patients. However, the model did not consider the value of the training, skills, knowledge and experience of the Medical Practitioner. Therefore, further research is required to come up with a consultation fee that is affordable and sustainable for both practitioners and patients.

## Introduction

Universal Health Coverage (UHC) mandates that individuals should have access to necessary health services without facing the risk of financial hardship or impoverishment [1]. This principle underscores the critical need for reasonable consultation fees that remain accessible to a broad population. In Zimbabwe, out-of-pocket payments, particularly for services like medical consultations, represent the primary source of health financing, accounting for over 39% of the total health financing structure [2].

Private healthcare providers in Zimbabwe often defend their pricing structures as a reflection of the operational costs associated with maintaining medical practices [3]. These costs encompass a range of essential expenditures, including registration fees, medical equipment and consumables, personnel salaries, and ongoing operational costs for utilities such as water and electricity [4].

This study addressed a significant knowledge gap, as no previous costing analysis had been conducted in Zimbabwe to validate or justify the prevailing consultation fees being charged in the private health sector. Though a Tariff Setting Committee is present in Zimbabwe [5], the formula for setting up these tariffs remains unknown, which may mean it is not informed by any scientific basis. There thus remains a paucity of information in this area which warrants research. In order for us to come up with a rationale price of consultation fees, there is need for a costing study to look at the various costing inputs and their implications on the pricing of health services. In developing a pricing model for medical consultations, the fee paid for consultation by the patient was the particular outcome measure of interest relative to the various input costs.

It was important to come up with a pricing model for consultation fees in order to have an ideal/actual price for those services. Only then could we ascertain whether costs were being minimized, that is resources were being allocated efficiently. Without an ideal/actual price for the healthcare services, sustainability would be compromised, as patients might be led to spend beyond the cost they can afford resulting in catastrophic spending or impoverishment. We therefore conducted the study to develop a Pricing Model for private consulting rooms that predicts the ideal price that could be charged for medical consultation services.

## Materials and methods

### Study design

An analytical cross-sectional study was utilized in which data was collected for the input costs of operating a Surgery as well as the consultation fee charged in the month of June 2023.

### Study setting

The study setting were general medical private consulting rooms in Harare, Zimbabwe. A total of 170 general medical practitioners responded to the survey through the designed online questionnaire administered using google forms. The four strata from which the surgeries were drawn include- Low, Medium and High Density Suburbs and the Central Business District. A combination of stratified sampling and simple random sampling was utilised to draw surgeries from each strata into the sample from the HPAZ database (sampling frame). The surgeries to be drawn from each strata into the sample, will be proportionate to the number of surgeries situated in that particular strata with respect to all the surgeries in Harare, i.e., CBD (37%), Low Density (18%), Medium Density suburb (21%), High Density suburb (24%). Thus forty respondents were from the High density suburb, 63 from the Central Business district, 31 Low density suburb and 36 from the medium density suburb.

### Study participants

The study participants were private general medical practitioners which are registered with the Medical and Dental Practitioners of Zimbabwe and Health Professions Authority Zimbabwe.

### Inclusion and exclusion criteria

The inclusion criteria for the study specified that general private consultation rooms located in Harare, which do not admit patients but provide general consultation services, were eligible. These rooms must be operational and have renewed their operating licenses for 2023, ensuring they are accessible to the general population seeking consultation services. Additionally, they should demonstrate consistent service provision, guaranteeing reliability in pricing. Conversely, the exclusion criteria ruled out public healthcare facilities, such as publicly funded or government-run clinics offering general consultation services. Private consultation rooms specializing in specific fields like cardiology or neurology were also excluded, along with those that do not accept out-of-pocket payments. Furthermore, consulting rooms affiliated with research or academic institutions focused primarily on teaching or research activities were not considered, nor were any consulting rooms that were permanently closed or temporarily unavailable during the study period in 2023.

### Sample size calculation and sampling method

The Dobson formula was used to calculate the sample size to ensure adequate representation of the population.
The Dobson formula used to calculate was as follows:-

$$n = \frac{Z^2 \ \times \ \alpha^2}{\Delta^2}$$

Where the variables are as follows:-
n = sample size
Z = 1.96 (Z-score corresponding to 95% level of significance)
α = 13 (estimated population standard deviation of pricing model variables; [6]
$\Delta$ = 2 (desired level of precision +/- 2units)

The calculation was thus as follows:

$$n = \frac{1.96^2 \ \times \ 13^2}{2^2} = 162$$

The sample size was adjusted factoring in 90% response rate. 162/0.9 = 180; therefore the adjusted **sample size = 180 (general medical consulting rooms)**

A combination of stratified sampling and simple random sampling were utilised. The four strata from which the surgeries were drawn into the sample include; Low, Medium and High Density Suburbs and the CBD. From these strata the surgeries were then drawn into the sample through simple random sampling from the HPAZ database (sampling frame).

The surgeries drawn from each strata into the sample, were proportionate to the number of surgeries situated in that particular strata with respect to all the surgeries in Harare, i.e., CBD (37%), Low Density (18%), Medium Density suburb (21%), High Density suburb (24%). This implied that 66 surgeries were drawn from the CBD into the sample, 33 from Low density, 37 from Medium density and 44 from High density to give a total sample size of 180 surgeries.

**Study variables**

**Independent variables**: Registration fees, cost of medical equipment, salaries, supplies and consumables, rentals, utilities, number of patients seen and profit margin.

**Dependent variable**: Price of general medical consultation services.

**Outcome**: Developing of a Pricing model for general medical consultation services.

The Multiple Linear Regression Equation will be as follows:-

$$\textit{Consultation Fee } (y) = \beta_0 + \beta_1 x_1 + \beta_2 x_2 + \beta_3 x_3 + \beta_4 x_4 + \beta_5 x_5 + \beta_6 x_6 + \beta_7 x_7 + \beta_8 x_8$$

Where:-

**Dependent variable**

$y$ = Consultation Fee

**Independent variables** -From Literature [7] & [8] and from experts.

$x_1$ = Registration Fees
$x_2$ = Cost of medical equipment
$x_3$ = Salaries
$x_4$ = Supplies and consumables
$x_5$ = Rentals
$x_6$ = Utilities, e.g., water, electricity etc.
$x_7$ = Number of patients seen
$x_8$ = Profit Margin
$\beta_0$ = Constant

$\beta_1$ to $\beta_8$ are the regression coefficients associated with the independent variables $x_1$ to $x_8$ respectively. The above model enabled the prediction of the ideal Consultation Fee that should be charged at a Surgery given the set of cost inputs which include; registration fees, salaries, consumables etc.

## Data collection tools and procedure

A questionnaire was developed to ascertain the various costing inputs required to operate a consulting room in Harare, as well as the consultation fee being charged. The questionnaire or costing form enabled the itemization, measurement and valuation of the various resources required to provide consultation services. In other words, the methodology was based on a detailed observation of resources required to operate private consultation rooms with the specific service output being GP consultation.

Data was collected from the practitioners in charge of the private consultation rooms through a questionnaire after obtaining their informed consent. The information collected included the consultation fee, which serves as the source of revenue for the doctors, it also served as the specific outcome measure of interest in the costing study relative to the various cost inputs which include registration/renewal fees, equipment costs, salaries etc. The consulting rooms included in the sample were obtained using a simple random sampling framework, from the HPAZ database which served as the sampling frame. The HPAZ database contains the names and addresses of all private health institutions in Zimbabwe [9]. The questionnaire was disseminated electronically through the use of Google forms©, and the responses collated in a Microsoft Excel© spreadsheet. The recruitment period for this study was from 13 September 2023, to 29 September 2023, during which 170 medical practitioners completed the online questionnaire. All costs were reported in United States Dollars (USD), as this is the primary trading currency for the private health sector in Zimbabwe. Costs were anchored to June 2023 values obtained through the questionnaire.

Pretesting of the questionnaire or costing form was done at consultation rooms in Harare CBD. The pre-test was done to check the feasibility of the questionnaire and to see if it was understood by the practitioners in the private consultation rooms. The FGD guide was also pre-tested for promptness with respect to saving time, since doctors are usually pressured for time due to their busy schedules. The pretesting of the research tools revealed satisfactory results, as the questionnaire and FGD guide were both effective in eliciting relevant data from participants.

## Data processing and analysis

Analysis was conducted using the statistical software Stata (Version 14). Multiple linear regression was utilized as the statistical model for the investigation of the mean change in the consultation fee (response) corresponding to a unit change in the cost of the inputs (explanatory variables). The ultimate objective of the regression analysis was to predict the consulting fee associated with fixed inputs to be costed in the study. Predictor variables which had p – values which were greater than or equal to 0.15 were excluded from the model.

To ensure the validity and reliability of the multiple linear regression model, the following assumptions were tested; linear relationship between the outcome variable (consultation fee) and the independent variables (rental, salaries, cost of consumables and equipment etc.), the independent variables should not be highly correlated with each other, residuals were to be normally distributed and homoscedasticity – the value of the error terms must be constant

T-tests were also done to ascertain if there is a linear relationship between the explanatory variables and the response variable (consultation fee). Hypotheses was laid out for one explanatory variable at a time in relation to its linear relationship with the response, whilst controlling for the other explanatory variables. The assumptions were found to be normal

Market prices for the medical equipment were used to determine their cost, which was then annuitized based on the equipment's life expectancy. This approach was crucial because the equipment's lifespan extended beyond the costing year (2023). Therefore, the cost was distributed over a five-year period, as the average life expectancy for medical equipment is five years [10].

To address variations in how practices reported their consumables, a standardized costing approach was adopted. Once a practice indicated possession or use of key equipment or service elements such as a dressing trolley, oxygen, or emergency medicines, it was assumed that related consumables including gloves, masks, syringes, linen savers, tongue depressors, cleaning materials, liquid soap, and paper towels, were part of the broader service package. These

items were then costed uniformly across all such practices to maintain consistency and comparability in the estimation of input costs. This methodological assumption was necessary to address potential underreporting or itemization gaps while ensuring a realistic and standardized input cost base.

The actual profit/loss made was obtained by subtracting the total running costs of the surgeries from their total annual revenue. The acceptable profit that should be made by surgeries was obtained by calculating 20% of the overall running costs for each of the surgeries. This aligns with common cost-plus pricing practices observed in health service delivery and reflects a conservative profit margin in private health care settings. Private healthcare providers in low-middle income countries typically operate at 15–25% profit margins to sustain operations and reinvest in services [11].

Analysis of Variance (ANOVA) was used to test if there is a statistically significant difference in the means of the consultation fees, and how the means varied among the four strata (CBD, Low Density, High Density and High density suburbs).

Statistical tests were conducted to ascertain if there was a statistically significant difference between the actual profit made by surgeries and the acceptable profit they ought to be making.

### Permissions and ethical considerations

Approval was sought and obtained from Joint Parirenyatwa Hospital and University of Zimbabwe's College of Health Sciences Research Committee (JREC), Medical and Dental Practitioners Council of Zimbabwe (MDPCZ) and Ministry of Health and Child Care.

Written consent was obtained from the participants before responding to the survey, privacy and confidentiality was assured to the participants by not writing names on the questionnaires, rather identifier numbers were utilized. Participants were given the option to opt out when they did not feel comfortable to continue.

## Results

### Characteristics of general practices

A total of 180 private consulting rooms registered with the Health Professions Authority of Zimbabwe (HPAZ) were invited to participate in the study. The study questionnaires were placed in google forms format and administered via email using the HPAZ database, which is routinely updated and contains accurate email addresses for licensed practices. Of the 180 practices, 170 completed the questionnaire, yielding a response rate of 94.4%. The study consisted of forty (40) respondents from the high-density suburb, 63 from the Central Business District, 31 low-density suburb and 36 from the medium density suburb. Thirty-three (19%) of the respondents were female and 137 (81%) were male. Their average years of working experience were noted to be 10.42 years. The high response rate is attributed to the study's endorsement by the Medical and Dental Practitioners Council of Zimbabwe, the short completion time of the questionnaire, and the pre-survey engagement efforts conducted with selected private sector representatives.

Table 1 reveals substantial geographic variation in practice characteristics, with CBD practices charging moderate fees (US$39 ± 9) while serving lower patient volumes (1617 ± 118), whereas low-density suburban practices commanded the highest consultation fees (US$49 ± 2) despite having the smallest patient base (1206 ± 130). High-density practices maintained the lowest fees (US$11 ± 2) while serving the largest patient volumes (3240 ± 231).

**Table 1. Annual Operational Characteristics and Costs (in 2023 US$) of General Practices by Location.**

| Characteristics of General Practices | Central Business District | High Density Suburb | Low Density Suburb | Medium Density Suburb |
|---|---|---|---|---|
| Location | 63 | 40 | 31 | 36 |
| Consultation Fee (Mean ± SD) | US$39 ± 9 | US$11 ± 2 | US$49 ± 2 | US$21 ± 2 |
| Patient Volume (Mean ± SD) | 1617 ± 118 | 3240 ± 231 | 1206 ± 130 | 2650 ± 89 |
| Cost of Equipment (Mean ± SD) | US$447 ± 33 | US$317 ± 16 | US$574 ± 51 | US$366 ± 12 |

## Cost of running a general practice

Table 2 below presents the key operational costs associated with running a consulting room, highlighting factors that influence pricing models for general medical consultations. Key expenditures include registration fees, rental costs, consumables, equipment, staff salaries, and utilities. The table also includes metrics on patient volume, providing insight into the number of patients seen relative to these operational expenses. Additionally, it illustrates actual profit or loss outcomes, comparing these with the acceptable profit levels needed to sustain the practice. This breakdown offers a comprehensive view of the financial elements that impact consultation fees and the sustainability of private medical practices.

## Multiple linear regression predictive model

The predictor variables which met the multiple linear regression assumptions and were included in the model were consumables, salaries, utilities, number of patients seen by doctor and actual profit made by the consulting room.

As seen above in Table 3, the p-values for all the predictor variables is less than 0.15, thus qualifying them to fit into the multiple linear regression model (they are all statistically significant). The R-Squared has a value of 0.95, meaning that 95% of the variation in consultation fees can be explained by the predictor variables in the model.

**Table 2. Cost of running of running a general practice stratified by location. Suburb (Mean amount in 2023 US$).**

| Category | Low density (Mean±SD) | Medium density (Mean±SD) | High density (Mean±SD) | CBD (Mean±SD) | Overall (Mean±SD) |
|---|---|---|---|---|---|
| Registration fees | 1,023.00±0 | 1,023.00±0 | 1,023.00±0 | 1,023.00±0 | 1,023.00±0 |
| Rentals | 32,051.61±2687 | 14,466.67±2239 | 7,446.00±1866 | 24,114.29±3378 | 19,596.71±9160.98 |
| Equipment | 573.87±51.10 | 366.39±11.50 | 316.75±16.20 | 446.98±33.00 | 422.41±93.25 |
| Consumables | 5,202.58±415.00 | 1,870.00±71.00 | 1,399.50±1770.00 | 3,752.38±280.00 | 3,064.59±1429.18 |
| Salaries | 24,077.42±3421.00 | 17,333.33±1328.00 | 9,960.00±2744.00 | 17,942.86±2255.00 | 17,054.12±5226.50 |
| Utilities | 222.58±80.80 | 390.00±30.40 | 621.00±58.50 | 386.67±41.40 | 412.59±141.54 |
| Patients seen | 1,206.00±130.00 | 2,650.00±89.00 | 3,240.00±231.00 | 1,617.00±118.00 | 2,143.00±791.09 |
| Actual profit/loss | −4,176.87±9178.00 | 21,200.61±6402.00 | 15,323.75±6907.00 | 15,371.92±11513.00 | 13,030.12±12445.84 |
| Acceptable profit | 12,630.21±1120.00 | 7,089.88±216.00 | 4,153.25±767.00 | 9,533.24±964.00 | 8,314.68±3016.15 |

**Table 3. Multiple Linear Regression Analysis. A multiple linear regression model was developed with the consultation fee as the dependent variable, to examine the influence of key cost-related factors such as consumables, salaries, utility costs, number of patients seen, and actual profit.**

| Consultation fee | Coefficient | St.Err. | t-value | p-value | [95% Conf | Interval] | Sig |
|---|---|---|---|---|---|---|---|
| Consumables | 0.006 | .001 | 10.96 | 0 | .005 | .007 | *** |
| Salaries | 0.001 | 0 | 8.21 | 0 | .001 | .001 | *** |
| Utilities | 0.012 | .003 | 3.38 | .001 | .005 | .019 | *** |
| patients_seen | -0.008 | .001 | -7.63 | 0 | -.011 | -.006 | *** |
| Profit | 0.0004 | .001 | 18.79 | 0 | 0 | 0 | *** |
| Constant | 7.81 | 4.691 | 1.66 | .098 | -1.453 | 17.072 | * |
| Mean dependent var | | 30.529 | SD dependent var | | | 15.025 | |
| R-squared | | 0.950 | Number of observation | | | 170 | |
| F-test | | 935.877 | Prob > F | | | 0.000 | |

*** p<.01, ** p<.05, * p<.1.

## Developed multiple linear regression equation

Five variables which satisfied the assumptions of Multiple Linear Regression analysis were fitted into the model, namely; cost of consumables, salaries paid to employees, cost of utilities, number of patients seen and actual profit. The following equation was developed from the Multiple Linear Regression analysis:-

$$Consultation\ Fee = 7.81 + (0.006 * consumables) + (0.001 * salaries)$$
$$+ (0.012 * utilities) - (0.008 * number\ of\ patients) + (0.0004 * Profit)$$

*Where $r^2 = 0.95$;* meaning that 95% of the variation in consultation fees can be explained by the predictor variables in the model.

Specifically the Multiple Linear Regression analysis found that there is a $0.006 increase (+/-0.0006) in the mean consultation fee for every $1 increase in the cost of consumables. The mean cost of consultation fee increases with $0.001 (+/-0.00009) for every $1 increase in salaries. The mean cost of consultation fee increases with 0.012 (+/- 0.004) for every $1 increase in the cost of utilities. There is a $0.008 decrease (+/- 0.001) in the mean consultation fee for every additional patient seen by a medical practitioner at an institution. Every $1 increase in actual profit results in the mean cost of consultation fee increasing with $0.0004 (+/- 0.00003).

The developed pricing model, based on Multiple Linear Regression analysis, provides a robust framework for estimating consultation fees for general medical consultation services among private consulting rooms in Harare, Zimbabwe. By inputting the costs of consumables, utilities, salaries, profit, and the number of patients seen into the developed equation, practitioners can accurately estimate the optimal consultation fee. This model takes into account the complex interplay between these independent variables, allowing practitioners to make informed decisions about their pricing strategy.

The application of this model is particularly useful for planning purposes. By knowing the estimated consultation fee, practitioners can determine the minimum amount they need to charge to cover their costs and maintain profitability. This information can also be used to inform negotiations with medical aid societies and private insurance companies, ensuring that practitioners receive fair compensation for their services. Furthermore, the model can be used to evaluate the impact of changes in costs or patient volume on the optimal consultation fee, allowing practitioners to adapt their pricing strategy in response to changing market conditions.

Ascertaining the estimated ideal consultation fee that should be charged by general medical practitioners.

The costs in Table 4 above are presented in 2023 US$, and the estimated ideal consultation fee was calculated using the following formula:

$$= \frac{(\ Reg\ fees + Rental + Equipment + Consumables + Salaries + Utilities + Patients\ seen + Profit)]}{Number\ of\ patients\ seen\ in\ a\ year}$$

**Table 4. Summary of all the predictor variables without stratification.**

| Variable | Obs | Mean | Std. Dev. | Min | Max |
|---|---|---|---|---|---|
| Reg fees | 170 | 1023 | 0 | 1023 | 1023 |
| Rental | 170 | 19596.71 | 9160.9 | 2640 | 36000 |
| Equipment cost | 170 | 422.41 | 93.25 | 300 | 700 |
| Consumables | 170 | 3064.59 | 1429.18 | 1080 | 6000 |
| Salaries | 170 | 17054.12 | 5226.50 | 6000 | 28800 |
| Utilities | 170 | 412.59 | 141.54 | 120 | 720 |
| Patients seen | 170 | 2143 | 791 | 960 | 3600 |
| Actual Profit | 170 | 13030.12 | 12445.84 | −25933 | 39177 |
| Acceptable profit | 170 | 8314.68 | 3016.15 | 2524.60 | 14042.60 |

$$\frac{(\,1023 + 19597 + 422 + 3065 + 17054 + 413 + 8315)]}{2143}$$

$$= 23.28$$

Therefore, the estimated ideal average consultation fee that could be charged is US$23.28.

It is important to take note that the acceptable profit (20% of the operating costs) was used in calculating the ideal consultation fee. Using the acceptable profit ensures that the price determined covers all costs and generates a profit margin that maintains the financial sustainability of the medical practice, and aligns with customer expectations. It allows for a realistic and sustainable pricing model to be developed.

### Test for significant difference between the actual profit and acceptable profit

A t-test was done to test if there was a statistically significant difference between the means of the actual profit and acceptable profit which were 13030.12 and 8314.68 respectively. The null hypothesis being there is no statistically significant difference between the actual profit and the acceptable profit made by general practitioners who are offering consultation services in private consulting rooms in Harare.

It can be seen in Table 5 above that there is a statistically significant difference between actual profit made by surgeries and the acceptable profit they ought to be making. At the 5% significance level, we have sufficient evidence against the null hypothesis and conclude that the mean actual profit made by surgeries is different from the acceptable profit that ought to be made.

### One way ANOVA test of consultation fee among the four suburbs

Table 6 shows that at the 5% level of significance, there is sufficient evidence of a difference in at least one of the means of the consultation rooms among the four suburbs.

As seen above in Table 7, all the differences of the means of the consultation fees are statistically significant, having a p value less than 0.05.

The One-way ANOVA and Bonferroni tests revealed that there were statistically significant differences in consultation fees between suburbs. These findings imply that the pricing of general medical consultation services in Harare is not uniform and may be influenced by geographical and socioeconomic factors. The variation suggests potential disparities in access to healthcare services, where residents in certain suburbs may face relatively higher financial barriers to consulting a medical practitioner.

To explore potential factors influencing profitability among private general medical consultation practices, Chi-square tests were conducted to examine associations between profit levels and two key variables; number of patients seen per month and suburb density.

A variable termed 'profit ratio' was derived by dividing actual profit by acceptable profit. Based on this ratio, profit levels were categorised into three groups; Below Target Profit (<0.8), Within Target Profit (0.8–1.2), and Above Target Profit

**Table 5. Test for the difference in the means of actual profit and preferred profit.**

|  | Actual profit | Acceptable profit |
|---|---|---|
| Mean | 13030.12 | 8314.68 |
| Variance | 154898996.70 | 9097154.10 |
| Observations | 170 | 170 |
| Pearson correlation = -0.45     Degrees of freedom = 169     T Statistic = 4.38. |  |  |
| P – value = 0.00002     T Critical value (two tail) = 1.97. |  |  |

**Table 6. Summary and One way ANOVA test of consultation fee among the four locations.**

| Location | Summary of Consultation fee | | | | |
|---|---|---|---|---|---|
| | **Mean** | **Std. Dev.** | **Frequency** | | |
| Central Business District | 39.05 | 8.61 | 63 | | |
| High Density Suburb | 11.13 | 2.11 | 40 | | |
| Low Density Suburb | 48.87 | 2.13 | 31 | | |
| Medium Density Suburb | 21.39 | 2.27 | 36 | | |
| Total | 30.53 | 15.03 | 170 | | |
| | Analysis of Variance | | | | |
| Source | SS | df | MS | F | P – value |
| Between groups | 33069.08 | 3 | 11023.30 | 359.97 | 0.0000 |
| Within groups | 5083.27 | 166 | 30.62 | | |
| Total | 38152.35 | 169 | 225.75 | | |

**Table 7. Bonferroni test of statistical significance of the differences in the consultation fees being charged in the four difference suburbs.**

| Comparison of consultation by Location (Bonferroni) | | | |
|---|---|---|---|
| **Location** | **CBD** | **High Density** | **Low Density** |
| High Density Suburb | −27.92 | | |
| | 0.000 | | |
| Low Density Suburb | 9.82 | 37.74 | |
| | 0.000 | 0.000 | |
| Medium Density Suburb | −17.66 | 10.26 | −27.48 |
| | 0.000 | 0.000 | 0.000 |

(>1.2). These thresholds were guided by a 20% profit margin benchmark used in the study, in line with widely accepted financial best practices regarding sustainable profitability [11].

Similarly, the number of patients seen per month was divided into three tertiles to facilitate categorical analysis. These tertiles were based on the distribution of responses, with the low volume group representing the lowest third of patient counts (1st tertile), the medium volume group covering the middle third (2nd tertile), and the high volume group comprising the highest third (3rd tertile). The cut-off values for these groups corresponded to the 33rd and 66th percentiles of the overall distribution of patients seen.

Table 8 shows there is a statistically significant association between suburb density and profit group classification, as indicated by the Chi-square value of 106.75 with 6 degrees of freedom and a p-value less than 0.001.

Table 9 shows that there is a statistically significant association between the number of patients seen (categorized into low, medium, and high volume) and the profit category (low, within target, or high profit), with a Chi-square value of 83.89 and a p-value less than 0.001.

## Discussion

In Zimbabwe, there is no set rate for consultation fees as it varies relative to a number of explanatory variables. Eight variables were analyzed in the study, namely; registration fees, rental, cost of equipment, cost of consumables, salaries paid to employees, cost of utilities, number of patients seen and profit margin. There was no linear relationship between registration fees and consultation fees being charged by general medical practitioners since the fees paid were constant throughout the 170 surgeries which responded to the survey. Five out of the eight variables were fitted into the multiple

**Table 8. Chi-square test showing the relationship between Suburb Density and Profit level.**

| Suburb Density | Profit Category | | | |
| --- | --- | --- | --- | --- |
| | Below Target (<0.8) | Within Target (0.8–1.2) | Above Target (1.2) | Totals |
| Central Business District | 18 | 8 | 37 | 63 |
| High Density | 0 | 3 | 37 | 40 |
| Low Density | 29 | 2 | 0 | 31 |
| Medium Density | 0 | 0 | 36 | 36 |
| Totals | 47 | 13 | 110 | 170 |

Pearson chi = 106.7548    Degrees of freedom = 6    P-value = 0.000.

**Table 9. Chi-square test showing the relationship between Number of Patients seen and Profit level.**

| Three Quantiles of Patients Seen | Profit Category | | | |
| --- | --- | --- | --- | --- |
| | Below Target (<0.8) | Within Target (0.8–1.2) | Above Target (1.2) | Totals |
| Low Volume | 40 | 5 | 12 | 57 |
| Medium Volume | 7 | 5 | 49 | 61 |
| High Volume | 0 | 3 | 49 | 52 |
| Totals | 47 | 13 | 110 | 170 |

Pearson chi = 83.8869    Degrees of freedom = 4    P-value = 0.000.

linear regression model as they satisfied the assumptions of the regression. These include; cost of consumables, salaries paid to employees, cost of utilities, number of patients seen and actual profit made by a medical practice. A high correlation was noted of rental and equipment with most of the other independent variables. Thus rental and equipment were removed from the regression. While rentals are not included in the regression equation, their inclusion in the total cost calculation ensures the model reflects real-world practice economics. This approach balances statistical rigor with practical applicability. The developed model can predict the increase or decrease in the mean consultation fee for a unit change in the explanatory variables.

The developed pricing model for consultation fees will greatly assist those in positions of policy to know the ideal price for consultation services. They will be able to ascertain whether costs are being minimized, i.e., resources are being allocated efficiently. Without an actual price for the healthcare services, sustainability would be compromised, as patients might be led to spend beyond the cost they can afford resulting in catastrophic spending or impoverishment [12].

The costing of health services in relation to the consultation fees in private consultation rooms which was done through this research will also provide transparency to consumers with regards to the health care services being offered to them [13]. This will instill confidence in the private health care service delivery system, and it ensures accountability on the part of health providers as well [14].

The ideal consultation fee which was obtained through calculation and using empirical data from the 170 surgeries in the sample was US$23.28. This price of delivering the consultation service was reached by taking into account various operational costs, including overhead expenses, staff salaries, and the necessary resources or equipment to offer the general consultation services. The premise of setting a consultation fee within the range of US$20 to US$25 helps cover these overhead costs and ensures the sustainability of the practice. Our calculated consultation fee of $23.28 aligns with the benchmarking data from the Association of Health Funders of Zimbabwe [15], while remaining comparable to regional outpatient service cost estimates from WHO-CHOICE [16]. According to WHO-CHOICE estimates, the average cost for outpatient consultations in upper low-income to lower-middle-income countries ranges between USD 20 and USD 30 in urban settings when accounting for full economic costs, including infrastructure, personnel, and overheads.

A statistically significant difference was observed by means of an ANOVA test with regards to the consultation fee being charged among the four suburbs, i.e., Low density suburb, Medium Density suburb, High Density suburb and Central Business District. Private consulting rooms in low density suburbs were seen to charge higher consultation fees compared to those in high density suburbs due to several factors discussed below.

Typically, the cost of operating a private consulting room in a low density suburb is higher than in a high density suburb. Low density suburbs often have higher property rental and maintenance costs, as well as other overhead expenses. These increased costs are often passed on to patients through higher consultation fees. Low density suburbs are also often associated with higher-income households, which may have a greater ability to pay for healthcare services. Private consulting rooms in these areas may strategically set higher fees based on the assumption that patients residing in these suburbs have a higher willingness to pay.

Private consulting rooms in low density suburbs may offer a higher level of convenience and accessibility compared to those in high density areas. Patients may perceive the added convenience and ease of access as valuable and are willing to pay a premium for it. The higher fees charged may reflect the perceived convenience and exclusivity associated with these locations. Private consulting rooms in low-density suburbs may also face less direct competition compared to those in high density areas due to factors such as limited healthcare facilities, higher operating costs, or a concentration of healthcare practitioners in certain locations. As a result, private consulting rooms in low-density suburbs may have the advantage of setting higher fees without facing significant pressure from nearby competitors.

A T-test revealed that there was a statistically significant difference between the actual profit being made by the surgeries and the acceptable profit that ought to be made. Thus advocating for a reasonable tariff (i.e., US$23) might be in tune with the acceptable profit margin derived from 20% of the running costs.

It is important to note that while the actual profit reflects the current financial situation, the preferred or acceptable profit serves as a benchmark or goal for the desired level of profitability. Hence, the difference between the two measures can highlight whether the consulting rooms are meeting, exceeding, or falling short of their target profitability. There could be several reasons for the difference between the actual profit and the preferred or acceptable profit of 20% of total running costs in private consulting rooms. Consulting rooms that are able to generate profits higher than the acceptable target may have implemented efficient operational strategies. This could include streamlining processes, reducing wastage, optimizing resource utilization, and implementing cost-saving measures. If there is a high demand for their services, consulting rooms might have the opportunity to charge higher prices. This can result from having a strong reputation, specialized services, or a unique value proposition that attracts a larger customer base and allows for higher profit margins.

Our analysis revealed significant variation in profit levels across multiple dimensions. Geographic factors played a substantial role, with profit distribution differing significantly by suburb density ($\chi^2$ (6) =106.7548, p = 0.000). Practices in the high density suburb and central business district were more likely to achieve higher profit levels compared to those in the low density suburb. Additionally, operational scale emerged as a significant factor, with patient volume strongly associated with profit category ($\chi^2$ (4) =83.8869, p = 0.000. Higher-volume practices demonstrated greater likelihood of achieving target and high profit margins. These findings suggest that both locational characteristics and practice scale constitute important determinants of financial performance in healthcare delivery.

At all stages in the process of developing the model, focus was kept concerning the feasibility of any actions in terms not only of the key variables (input costs and subsequent consultation fees) but also of the broader political process. There is a need to ensure that the priorities set are feasible within the social and political climate and within the context of available resources concerning the developed pricing model for general medical consultation services.

In conclusion, the developed pricing model offers a valuable tool for practitioners to estimate optimal consultation fees for general medical consultation services. By inputting the costs of key independent variables into the developed equation, practitioners can make informed decisions about their pricing strategy, promote business sustainability, and contribute to the delivery of high-quality healthcare services.

## Strengths and limitations

This study provides significant strengths through its comprehensive mixed-methods analysis of the key cost drivers influencing private consultation fees in Zimbabwe. By rigorously testing assumptions and developing a context-specific pricing model ($R^2 = 0.95$), the study offers policymakers and practitioners an empirically grounded tool to standardize fees while accounting for clinic-level cost structures. The focus on Harare's private consulting rooms addresses a critical evidence gap in low-resource settings where pricing mechanisms are often unclear. The transparent methodology, including publicly shared questionnaires and raw data, further enhances the model's utility for decision-making.

However, several limitations should be considered. The geographic focus on Harare and temporal limitation to 2023 data may affect generalizability to rural practices or other time periods, suggesting need for broader future studies. While our fee model demonstrates strong predictive power, the profit comparison model's lower explanatory value ($R^2 = 0.21$) reflects expected clinic-level heterogeneity in operational efficiency and patient mix – a well-documented phenomenon in private healthcare economics [17]. The model also focuses on cost inputs and does not incorporate qualitative factors like practitioner expertise or patient-perceived value that may influence real-world pricing strategies.

Notwithstanding these limitations, this research provides the first robust pricing framework for Zimbabwe's private clinics, with immediate applicability for health financing policy. The findings establish a crucial baseline for future studies to build upon, particularly through incorporation of non-cost variables and expanded geographic sampling.

## Conclusion

In conclusion, this research study aimed to develop a pricing model for general medical consultation services, and determine the ideal consultation fee that could be charged by general medical doctors. Through the examination of 170 general private consultation rooms in Harare and their corresponding consultation fees, the findings from the study suggest that an optimal price for general private medical consultation services might be US$23. Given the observed heterogeneity across density and patient volume categories, future pricing policy frameworks may consider differentiated consultation fee benchmarks for distinct practice types.

Overall, this research study contributes to the existing body of knowledge in the healthcare economics field, shedding light on the relationship between the cost of inputs for operating private consultation rooms and the consultation fees charged. It provides a foundation for further research and decision-making in the healthcare industry. It also has implications for healthcare affordability, access, and financial sustainability for both practitioners and patients.

## Supporting information

**S1 Fig. Relationship between Estimated Actual Profits and Ideal Profits for General Medical Practices in Harare, Zimbabwe.** This scatterplot shows the relationship between estimated profits and ideal profits (from the predictive model) for 170 general medical practices in Harare, Zimbabwe.
(PDF)

**S1 Table. Correlation matrix of the independent variables and the outcome variable.** This matrix assesses multi-collinearity among cost variables, patient volume, profit, and the outcome variable (consultation fee).
(PDF)

**S2 Table. Correlation matrix with the five independent variables with minimal multi-collinearity.** This matrix presents the correlations among the variables retained after addressing multi-collinearity (consumables, salaries, utilities, number of patients seen, and profit), showing their relationship with the consultation fee.
(PDF)

**S3 Table. Comparison of Regression Models for Profitability vs. Consultation Fee Determination.** This table compares the regression models for profitability and the one for fee determination.
(PDF)

**S4 Table. Showing the distribution of number of patients seen across the three tertiles.** This table presents the number of patients seen per month, divided into three tertiles to facilitate categorical analysis.
(PDF)

**S1 File. Data collection questionnaire for private consulting rooms in Harare.** This questionnaire was used to collect data from general practitioners operating private consulting rooms.
(PDF)

**S2 File. Ethical Clearance Correspondence from Medical and Dental Practitioners Council of Zimbabwe.** This document provides the official communication from the national medical regulatory body which granted permission to conduct the study among its registered practitioners.
(PDF)

**S3 File. Ethical Clearance Letter from Joint Research Ethics Committee.** This document provides the formal ethical approval for the study.
(PDF)

## Acknowledgments

I would like to acknowledge the following for their unwavering support; Dr J. January for imparting a wealth of public health knowledge with regards to the challenges being faced in the Zimbabwean health sector. Health Professions Authority Zimbabwe for providing statistical information of all private health institutions in Zimbabwe. Doctors who assisted in the identification of the predictors and design of the questionnaire for assessing the cost of starting a medical surgery in Harare. Special thanks to my family and colleagues for the support that they gave me in developing this pricing model.

## Author contributions

**Conceptualization:** Chengetedzai Gota.

**Data curation:** Chengetedzai Gota.

**Formal analysis:** Chengetedzai Gota.

**Funding acquisition:** Chengetedzai Gota.

**Investigation:** Chengetedzai Gota.

**Methodology:** Chengetedzai Gota.

**Project administration:** Chengetedzai Gota.

**Resources:** Chengetedzai Gota.

**Software:** Chengetedzai Gota.

**Supervision:** Gibson Mandozana, Richard Makurumidze, Shepherd Shamu.

**Validation:** Chengetedzai Gota.

**Visualization:** Chengetedzai Gota.

**Writing – original draft:** Chengetedzai Gota.

**Writing – review & editing:** Gibson Mandozana, Richard Makurumidze, Shepherd Shamu.

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
