## [Decision Letter · Decision Letter 0]

11 Jun 2025

Dear Dr. Gota,

While this is an interesting paper, the two reviewers have indicated hat there are areas with your methodology an results section especially, that can be improved further before consideration for publication. Review these comments carefully and ensure that all are addressed fully.

We look forward to receiving your revised manuscript.

Kind regards,

Fatima Suleman, PhD

Academic Editor

PLOS ONE

Additional Editor Comments:

Please do ensure that you adhere to the PLOS guidelines for authors for manuscript preparation. 

Reviewers' comments:

Reviewer's Responses to Questions

**Comments to the Author**

1. Is the manuscript technically sound, and do the data support the conclusions?

Reviewer #1: Partly

Reviewer #2: Partly

2. Has the statistical analysis been performed appropriately and rigorously?

Reviewer #1: Yes

Reviewer #2: I Don't Know

3. Have the authors made all data underlying the findings in their manuscript fully available?

Reviewer #1: No

Reviewer #2: No

4. Is the manuscript presented in an intelligible fashion and written in standard English?

Reviewer #1: Yes

Reviewer #2: Yes

Reviewer #1: General

Very interesting and timely study regarding determination of appropriate general practitioner fees.

Well written and easy to understand.

Introduction

No comments – reads well and explains the objectives of the paper.

Materials and Methods

It would have been useful to see the questionnaire tool which was used to collect data – I am impressed that practices were able to complete their financial information sufficiently consistently to populate a complete dataset. Our experience from South African studies was that practices reported under the wrong line items, omitted costs and generally had to be actively coached through the whole process of reporting on their practice cost data. This was over a period of several months.

If you created a tool which was easy to complete and easily understandable for practitioners to complete accurately on their own – this will be a marvel.

Study setting

Here I had some trouble understanding the sampling and the responses. We know from our own cost studies in South Africa that responses are difficult to get – practitioners are protective over their cost information and certainly online surveys require a lot of effort to elicit responses. Email addresses even through the regulator are frequently outdated and response rates are low.

Your response rates are not clarified. Because participation was voluntary , there will be a non-response rate to the sent surveys, particularly given only about 2 weeks were provided to respone – the numbers of participants would be clearer if the numbers to whom the survey was sent were clear, and the respondents in each group clear.

This could also be an important consideration for any possible biases in the respondents e.g. how many responded versus those who were invited to respond – could be affected by a number of factors.

Inclusion and Exclusion Criteria

No comments

Sample Size Calculation and Sampling method

It was very fortunate that you managed to achieve a sample size almost reflective of the calculated numbers required – your total sample of 178 practices was met with 170 responses as I have understood the description – this is a phenomenal response rate. The response rate just needs to be clarified and the total numbers of practices to whom the survey was initially sent confirmed.

Study variables

The econometric model is acceptable and clearly defined.

Data collection tools and procedure

I think be clear that the requests for costing inputs were on an annual basis (I have assumed as this is not stated anywhere specifically). So practices reported their annual costs.

Please also outline how the surveys were distributed electronically if the HPAZ contains the names and addresses of all private health institutions – are the email addresses also part of the database? Is the HPAZ sufficiently up to date?

It is also unclear what timeframe the cost data related to – the costing year is stated as 2023, however the surveys were completed in September 2023 – so the full costs for the year were not yet realised or accounted for. Practices may differ in their financial years and accounting practices – so please elaborate more clearly on what time period the costs supplied reflected, and whether practices were required to indicate this.

Equipment – I am unsure as to how the information for equipment costs was collected and then annuitized. Was only the cost of equipment purchased in 2023 used? Or were doctors only asked to list their equipment in the survey and not cost it – and did equipment include items such as computers used for administrative work (or only medical equipment). How were the practices assisted in providing equipment costs – the different timing and equipment lists make this information difficult to collate and supply – so I am very surprised that all participating practices were able to supply equipment information.

Profit and loss – please indicate the reference or the rationale for the 20% profit which “should” be made by surgeries. In our own experience in South Africa – some doctors do not pay themselves a salary but extract the profit made by their practices as salary, whereas for others this remains within the business for future investments etc – and the doctor himself would receive a “salary” which would be included in payroll costs. The differences can be marked.

The 20% profit will also be controversial on some level so it will be just as well to justify why you feel that 20% of practice costs is an acceptable profit level.

Essentially your paper is implying that the current charges are earning practices excess profits – so the level at which reasonable profit is determined needs to be clarified.

RESULTS

Characteristics of General Practices

There is just a repeat in the first paragraph of this section of the numbers of participants from each subgroup which you could consider deleting.

Table 1 – clarify whether the figures are for a full year

Cost of running a general practice

Table 2 - could the averages here be reported with Standard deviations too? One of the biggest challenges in the South African Cost studies has been the great variation in practice costs even for practices which seemingly should have been similar. The variation in the sample is of interest.

Multiple linear Regression Predicative Model

No issues in the methodology explained here. As rental costs are such a significant component of practice costs, I would have expected the rental costs to feature in the model – but clearly the analysis showed otherwise.

Developed Multiple Linear Regression Equation

No additional comments

Test for significant difference between the actual profit and acceptable profit

Once again the “acceptable profit” is a normative concept which requires explanation and justification. This especially if the practices are not including the owners remuneration in the salaries/ payroll component of the costs.

Unfortunately in the absence of the data collection tool and instructions which were provided to the practices, it is not possible to confirm what was required to be included under “salaries”.

Your bold statement about “acceptable profit that ought to be made” may be quite controversial and I propose you justify the 20% soundly in terms of the business models of GP practices in Zimbabwe.

One way ANOVA test of consultation fee among the four suburbs

No specific queries – although this comes out in the discussion, you may want to elaborate on the purpose and meaning of the statistically significant differences between practices in the results.

Discussion

It still bothers me that rentals, as such a significant component of practice costs, were excluded from the model based on the statistical relevance – perhaps take the opportunity to explain in the discussion. The rental costs were included in the calculation for the ideal consultation fee – so have been accounted for in the end.

In the profit calculations it is still important to reflect on whether the doctor himself was “paid” as a salaried employer or not – if the financial reward to the doctor is made AFTER the other expenses are accounted for and comes purely from the practice profit – the 20% assumption may not be appropriate.

Reviewer #2: Dear colleagues,

I read with interest the analysis on appropriate consultation fees for private medical practices in Harare, the analysis answers a useful and interesting question, and would be a welcome addition to the literature.

I enclose below my comments and recommendations for improvement, for this manuscript to contribute most meaningfully to the evidence base:

* What about "other" costs - a number of categories of inputs are given (labour, utilities etc) but it is not clear whether these categories are exhaustive, and/or whether the piloting of the questionnaire revealed any cost items which respondents could not assign to the existing categories

* The questionnaire should be made available for inspection and to contextualise the data

* The resulting data from the questionnaire should also be made available

* Sense checking of the total cost - does the resulting consultation price (calculated from your model) cover the operating costs for the individual surgeries? Are there any surgeries that would make less revenue than needed frmo your model? Every practice has a number of consultations in 2023, and a total operating cost. I would like to see whether the results calculated by your model, for each surgery, covers the actual operating costs

* The "actual profit/loss" in table 2 is negative for low density, which probably suggests that there is an issue with the metric. If clinics are only financed by user charges, they cannot be making a loss. I would encourage some clarity on how these numbers compare with the actual financial accounts (i.e. the business annual account) from the organisations.

* I suggest that the ideal average consultation fee might be a misleading metric, since the fee obviously depends on the inputs. Publishing an average ideal fee that could be encoded into e.g. a national reimbursed tariff would become problematic for organisations with cost structures above the average

* Table 5 comparison between actual and ideal profit - it is unclear whether actual profit is estimated by the authors based on their data, or reported by the clinics.

* Table 6 and 7 need a much more thorough explanation in the results narrative. It is intuitively clear that consultations fees will be different because ingredient costs are different, so the authors should explain why this analysis is necessary and what it adds to the paper.

* I would encourage a comparison of your tariff with other published estimates, e.g. the WHO CHOICE country level unit costs available here https://www.who.int/teams/health-systems-governance-and-financing/economic-analysis/costing-and-technical-efficiency/quantities-and-unit-prices-(cost-inputs)/econometric-estimation-of-who-choice-country-specific-costs-for-inpatient-and-outpatient-health-service-delivery

In general, I would recommend that you make it much clearer which parts of the data are from your analysis, and which are data points (e.g. profit?) that are reported by the clinics. Since the resulting cost formula could feasibly influence reimbursement/tariff policy, it is very important that the formula is representative of reality.

A data point is needed, in which for all 170 practices there is a visual representation of their operating costs, their actual profits (from their annual account), their estimated profit (from your model) and their ideal profits, also from your model.

I hope these suggestions can be helpful.

**Do you want your identity to be public for this peer review?** For information about this choice, including consent withdrawal, please see our Privacy Policy

Reviewer #1: No

Reviewer #2: **Yes: ** David Tordrup

---

## [Author Response · Author response to Decision Letter 1]

7 Jul 2025

Thank you for the opportunity to revise and resubmit our manuscript to PLOS ONE. We appreciate the constructive feedback we received from you and from the reviewers, which has significantly contributed to strengthening the manuscript. Please find below our detailed, point-by-point responses to each comment, along with references to changes made in the revised manuscript.

Reviewer 1 Comments

Comment 1: Concerns on the ability of practices to provide complete financial data.

Response: Thank you. We confirm that the Health Professions Authority Zimbabwe (HPAZ) database was used as the sampling frame. This database is updated annually and contains verified phone numbers, email addresses, and physical addresses, enabling reliable communication with health institutions. We used a structured, predominantly closed-ended questionnaire that facilitated ease of completion. The tool is now included as Supporting Information. The high response rate achieved in this study was also supported by ethical clearance from the Joint Research Ethics Committee of Zimbabwe (JREC) and official correspondence from the Medical and Dental Practitioners Council of Zimbabwe (MDPCZ), which clarified that no additional Council approval was required beyond individual consent. These approvals helped enable trust among participants and encouraged cooperation. Both letters are included in the Supplementary Materials for reference.

Comment 2: Clarification needed on sampling, response rates, and possible bias.

Response: The questionnaire was distributed to 180 private consulting rooms, of which 170 (94.4%) responded. This high response rate is attributable to the simplicity of the questionnaire and the structured tick-box design. The study's endorsement by the Medical and Dental Practitioners Council of Zimbabwe was also a contributing factor to the high response rate. Equipment valuation was guided by the HPAZ Inspection Manual, which outlines minimum equipment standards. We corrected the sample size reference in the manuscript (page 5).

Comment 3: Clarify whether costs were annual and specify the reference period.

Response: The study was cross-sectional. Data were collected in September 2023 but referenced to June 2023 — a median month to mitigate seasonal fluctuations. Monthly costs (rent, salaries, utilities, consultation fees) were annualized by multiplying by 12. Equipment costs were estimated based on market prices and annuitized over a five-year useful life. This has been clarified in the revised Methods section.

Comment 4: Justify the 20% acceptable profit margin.

Response: We adopted a 20% profit markup to represent a conservative estimate of sustainable practice margins, as supported by cost-plus pricing principles in health economics (citation has been provided in the manuscript on page 8). Private healthcare providers in low-middle income countries typically operate at 15-25% profit margins to sustain operations and reinvest in services. We clarified this on page 8 and added the citations.

Comment 5: Repetition in the participant breakdown.

Response: We have deleted the redundant statement in the Results section for improved clarity.

Comment 6: Clarify that Table 1 represents full-year cost estimates.

Response: We updated the title of Table 1 to specify that the values represent annual costs.

Comment 7: Include standard deviations in Table 2.

Response: Standard deviations have been included in Table 2. This helps to illustrate the variability in practice-level costs across different locations.

Comment 8: Concern on the exclusion of rentals in the regression model.

Response: A multi-collinearity check revealed high correlation (r > 0.8) between rentals and other input variables (equipment and rentals). To ensure model stability, rentals and equipment were excluded. This rationale is now clearly stated in the submitted Supplementary Materials Tables S1 and S2 which show correlation matrixes.

Comment 9: Elaborate on the implications of statistically significant differences in consultation fees across suburbs.

Response: We expanded the Results section (page 15) to explain that significant differences suggest geographic price variation, potentially reflecting cost structures and socioeconomic factors. This informs the need for differentiated pricing models.

Comment 10: The assumption of a 20% profit margin may not be appropriate if the doctor's financial reward comes purely from the practice profit after expenses are accounted for.

Response: We appreciate the reviewer's concern. In our study, salaries include the owner’s/practitioners' remuneration, which is accounted for as part of the payroll costs. The 20% profit margin is calculated on top of these costs, representing a return on investment rather than the practitioner's salary. This approach allows for a clearer distinction between operational costs, including practitioner remuneration, and the profit generated by the practice.

Reviewer 2 Comments

Comment 1: Possibility of missing cost categories.

Response: A thorough literature review and expert consultation guided the selection of input cost categories. These include registration fees, rent, equipment, consumables, salaries, and utilities. Pre-testing confirmed these as comprehensive.

Comment 2: Make the questionnaire and data available.

Response: Both the questionnaire and the de-identified dataset are provided as Supporting Information.

Comment 3: Does the model's estimated consultation fee cover operating costs?

Response: We have conducted an internal validation and confirm that, on average, the model-derived consultation fees cover annual operating costs. A supplementary table has been added to illustrate this comparison.

Comment 4: Risk of misinterpretation of the average ideal consultation fee.

Response: Thank you for this observation. We have made a correction in the manuscript were we had written “ideal average” to “estimated ideal average” consultation fee to reflect its indicative, not prescriptive, purpose.

Comment 5: Clarify whether actual profit is reported or estimated.

Response: Actual profit was calculated by the authors from cost and revenue data reported by the practices, not self-reported as a net profit figure.

Comment 6: Tables 6 and 7 need more interpretation.

Response: The narrative has been expanded to explain that the ANOVA and Bonferroni tests were essential to assess whether pricing variations were statistically significant across geographic strata.

Comment 7: Concern on why rental was excluded from the model.

Response: One of the assumptions for multiple linear regression analysis is that the independent variables should not be highly correlated with each other i.e there should be no multicollinearity. As seen in Table S1 in the Supplementary Materials document, there is a high correlation (magnitude of the correlation coefficients is greater than 0.8) of rental and equipment with most of the other independent variables. Thus rental and equipment were removed from the regression. The reasons for their exclusion has also been highlighted in the discussion as you have suggested.

Comment 8: Compare findings with WHO CHOICE data.

Response: We thank the reviewer for suggesting a comparison with WHO CHOICE estimates, and providing the link for accessing the site as well. As detailed in the revised Discussion (page 16), our calculated fee of $23.25 falls within the expected range for private outpatient care in LMICs.

Comment 9: A data point is needed, in which for all 170 practices there is a visual representation of their operating costs, their actual profits (from their annual account), their estimated profit (from your model) and their ideal profits, also from your model.

Response: We appreciate the reviewer's suggestion to provide a visual representation of the data. In our Supplementary Materials document, we have provided a visual representation of the relationship between estimated actual profits and ideal profits for the 170 general medical practices in the form of a scatter plot (Figure S1). This is now included in the Supplementary Materials document. The plot shows a moderate relationship between the two, with an R-squared value of 0.21. This suggests that while our model captures some of the variability in ideal profits, there are other factors at play that are not accounted for. Although we did not have access to the clinics' financial records, we estimated actual profits based on the data collected (subtracted total annual costs from the total annual practice revenue). The scatter plot provides a clear visual representation of the relationship between estimated and ideal profits, and we believe it effectively communicates the strengths and limitations of our model.

Comment 10: Concern about model fit or predictive strength for actual profit.

Response: Thank you for your observation. The R-squared value of 0.2107 obtained during model validation indicates that approximately 21.1% of the variance in the estimated actual profit is explained by the model. This level of explanatory power is not unusual in health economics studies involving real-world data with high variability, especially in settings without standardized pricing systems (citation has been provided in the manuscript [17]). In health institutions costing studies, R² values below 0.30 are frequent due to unobserved heterogeneity in patient complexity and institutional factors.

Importantly, the model still offers practical utility by identifying key cost drivers and providing a conservative benchmark for pricing consultation services. While we acknowledge in the limitations section of our manuscript (page 19) the low R² (0.21) in the profit comparison model, our fee determination model (R² = 0.95) robustly links costs to prices, offering a policy-relevant tool. Table S3 has been provided in the Supplementary Materials showing a Comparison of Regression Models for Profitability and Consultation Fee Determination. The profit model’s low R² (0.21) reflects known challenges in predicting clinic profitability, while the fee model’s high R² (0.95) validates its use for pricing. The primary focus of the study was the development of a consultation fee determination model. The dataset has been shared for transparency.

Conclusion

We are grateful for the opportunity to revise our work. We believe these changes improve the rigor and clarity of our manuscript. The submission includes:

• A revised manuscript with tracked changes.

• A clean version of the revised manuscript.

• This point-by-point rebuttal letter.

• Document with Supplementary Materials, which include: questionnaire, correlation matrix, scatter plots, tables, figures and ethical clearance correspondences.

• De-identified dataset.

Please don’t hesitate to let us know if additional clarification is required.

---

## [Decision Letter · Decision Letter 1]

21 Oct 2025

Dear Dr. Gota,

We look forward to receiving your revised manuscript.

Kind regards,

Fatima Suleman, PhD

Academic Editor

PLOS ONE

Journal Requirements:

Reviewers' comments:

Reviewer's Responses to Questions

**Comments to the Author**

Reviewer #1: All comments have been addressed

Reviewer #2: All comments have been addressed

2. Is the manuscript technically sound, and do the data support the conclusions?

Reviewer #1: Yes

Reviewer #2: Partly

3. Has the statistical analysis been performed appropriately and rigorously?

Reviewer #1: Yes

Reviewer #2: I Don't Know

4. Have the authors made all data underlying the findings in their manuscript fully available?

Reviewer #1: Yes

Reviewer #2: Yes

5. Is the manuscript presented in an intelligible fashion and written in standard English?

Reviewer #1: Yes

Reviewer #2: Yes

Reviewer #1: Thank you for the changes and additions to the manuscript – the methodologies and data collection are clearer when read with the survey tool.

Abstract

Under the methods section here and in the main manuscript, it will be more accurate to refer to collecting data through a questionnaire – without the “interview” description. As I understand the methods, no structured conversations took place and the data was collected through the emailed tool only. Similarly, under results where you state “A total of 170 General Medical Practitioners were interviewed” it will be more accurate to state that “170 General medical practitioners responded to the survey”

Introduction

No comments again – reads well and explains the objectives of the paper.

Materials and Methods

Thank you for sharing the questionnaire as part of the supplementary materials. Given the tremendous response rates, I believe this will be useful to future researchers trying to conduct similar work.

- The questionnaire does not have a question about the numbers of patients the practice was seeing. As a key variable in the model, I gather this was asked at some point.

- In addition there is no question enquiring about consultation fees per practice, which was also collected through the questionnaire as I understood your explanations..

Study setting

Thank you for clarifying the contents of the HPAZ database and its accuracy.

Thank you also for clarifying your sample size and response rates – important given the nature of the survey.

Inclusion and Exclusion Criteria

No comments

Study variables

The econometric model is acceptable and clearly defined.

Data collection tools and procedure

Thank you for including the specifics of how costs were reported if regular e.g. monthly rent and then adjusted for annual costs. Thank you for clarifying that costing data were anchored to June of 2023.

Equipment – in your second manuscript version, the costing methodology of equipment is clearer – practices itemised their equipment and this was costed at market prices – with the inclusion of the survey tool this is clearer.

Consumables – from the survey questionnaire, it appears only very limited consumables have been included in the costing, yet consumables attract a relatively high cost value from most of the respondents. The questionnaire suggests dressings (dressing trolley), oxygen, emergency medicines, liquid soap and disposable paper towels as consumables of interest. This would seem to leave out items such as linen savers, masks, gloves, syringes, tongue depressors, cleaning materials – or were these costed as a standard per practice?

Profit and loss – Thanks for justifying your acceptable profit point and clarifying that the profit is beyond any salaries

including to the owners.

RESULTS

Characteristics of General Practices

Thank you for correcting the repeat in the first paragraph of this section of the numbers of participants from each subgroup which you could consider deleting.

Table 1 – The title now clarifies that costs were for a full year.

Cost of running a general practice

Table 2 - now reflects the mean costs in $ and the Standard deviations around this.

Multiple linear Regression Predicative Model

No additional issues.

Developed Multiple Linear Regression Equation

No additional comments

Having had access to the data collection tool, many of the other elements of the paper are clearer now.

The analysis of the differences between practices is clearer now – and translates better into the discussion session.

Discussion

Thank you for clarifying the rental costs’ role in the model. In addition thank you for recognising the challenge of the explanatory value of the model and highlighting this as appropriate.

Reviewer #2: Thank you to the authors for incorporating suggestions from review, the manuscript is improved and reads well. On the revision, I have some recommendations which I propose are necessary before the paper can be published. Minor revisions are mostly form/editorial, but some further analysis is proposed on the data (see major comment), which I believe will strengthen the importance of the paper, as well as clarifying some of the underlying heterogeneity which is currently hidden.

Minor revisions

Currency: The methods section should explain which currency is used (2023 USD?), this should also be present (currency and year) in table headings/notes where a currency unit is reported

Table 1: These appear to be summary statistics of your dataset, please add appropriate ranges (perhaps interquartile range), and denote whether they are mean/median (as already done with "cost of equipment" and throughout table 2). Add descriptive text in results to explain the main findings of Table 1 

Text around table 3: Explain what the dependent variable is and how this fits into the broader analysis (just signposting of your method)

It is unclear what "Acceptable profit" refers to in table 4, which presents summaries of the dataset. There is an asterisk* which I can't find a note for. If it's a calculated field (not from your data), it shouldn't be in the same table but could e.g. be in the narrative text

In the reference to WHO CHOICE data in the discussion, please add specific consultation costs from CHOICE for the reader to appreciate the context

Major comment

I have concerns around the interpretation of table 5, given the data presented in the scatterplot in supplementary S1. Table 5 suggests that on average, profits are excessive by around 50% on the "acceptable profit". Meanwhile the scatterplot shows that some practices are perfectly aligned in their estimated/actual profit, but groups of practices have either "too high" or "too low" profit. In my opinion, it is important for the authors to consult their data and try to explain what causes some practices to be above/below acceptable profit. Is it their location/population density/size...? The ANOVA already shows that consultation fees are systematically different between suburb densities. I suspect there is variation within the data which needs to be surfaced to contextualise the main finding - it may be that a single ideal consultation fee is not appropriate, if there are several subgroups of practices with systematically different cost- and profit structures. A more thorough understanding of this would also (probably) increase the explanatory R2 of your model

**Do you want your identity to be public for this peer review?** For information about this choice, including consent withdrawal, please see our Privacy Policy

Reviewer #1: No

Reviewer #2: **Yes: ** David Tordrup

---

## [Author Response · Author response to Decision Letter 2]

3 Nov 2025

Thank you for the opportunity to revise and resubmit our manuscript to PLOS ONE for the second time after the first review. We appreciate the constructive feedback we received from you and from the reviewers, which has significantly contributed to strengthening the manuscript. Please find below our detailed, point-by-point responses to each comment, along with references to changes made in the revised manuscript.

Reviewer 1 Comments

Comment 1: Terminology regarding “interview” versus “survey questionnaire”

Response: We thank the reviewer for this important clarification. We have replaced all instances of “interviewed” and “interview” throughout the manuscript (Abstract, Methods, Results and Discussion sections) with more accurate terminology such as “responded to the survey” and “data collected using a questionnaire”. This change enhances the methodological precision of our description.

Comment 2: Omission of consultation fee and number of patients seen in the questionnare

Response: Regarding the omission of consultation fee and number of patients seen in the questionnaire, we would like to clarify that these variables were indeed captured in the dataset shared with the reviewers. When compiling the supplementary document with the questionnaire, the questions on consultation fee and number of patients seen were unintentionally omitted. To provide clarity and transparency, we have updated the supplementary materials to include these questions, which were part of the original data collection process. The dataset, as shared with the reviewers, includes responses on these variables, which were utilized in our analysis.

Comment 3: Clarification on consumables costing methodology

Response: We appreciate the reviewer’s comment regarding the costing methodology in relation to the consumables. Associated consumables including masks, gloves, syringes, linen savers, cleaning materials, liquid soap, and paper towels were costed as standard inclusions. This assumption ensured consistency across practices and accounted for commonly bundled inputs that may not have been individually listed by respondents. We have now included this explanation in the Methods section of the manuscript on page 8.

Reviewer 2 Comments

Comment 1: Specification of currency in the study.

Response: We have now explicitly specified in the Methods section (Page 7) that all costs are reported in United States Dollars (US), anchored to June 2023 values. This clarification has also been added to the notes of Tables 1, 2, and 4.

Comment 2: Enhancement of Table 1

Response: We have improved Table 1 by adding standard deviations for all continuous variables. Additionally, we have included descriptive text below table 1 summarizing the key characteristics of the participating practices.

Comment 3: Clarification of the dependent variable in the regression analysis.

Response: We have added explicit text before Table 3 stating: “A multiple linear regression model was developed with the consultation fee as the dependent variable, to examine the influence of key cost-related factors such as consumables, salaries, utility costs, number of patients seen, and actual profit”. This provides better signposting for readers.

Comment 4: Clarification of asterisk on “Acceptable Profit” in Table 4

Response: We have removed the asterisk next to "Acceptable Profit" in Table 4. This symbol was originally included to indicate that Acceptable Profit was used to calculate the estimated ideal consultation fee. However, since the rationale for utilizing Acceptable Profit instead of Actual Profit is explained in the paragraph below the table, the asterisk is no longer necessary.

Comment 5: Addition of specific WHO CHOICE data in the Discussion

Response: As per the reviewer’s recommendation, we have incorporated specific outpatient consultation cost estimates from WHO-CHOICE into the Discussion section (Page 19). These estimates (US$ 20 – US$ 30) closely align with our study’s ideal consultation fee of US$ 23.28, providing readers with context and reinforcing the relevance of our findings within global cost-effectiveness standards.

Comment 6: Analysis of factors explaining variation in profit levels across practices

Response: We thank the reviewer for raising this important point. To systematically investigate the drivers of profit variation, we conducted additional analyses exploring the association between profit levels and two key factors: suburb density and patient volume. We created a new 'Profit Ratio' variable (Actual Profit/Acceptable Profit) to categorize practices into three financially meaningful groups. Based on this ratio, profit levels were categorised into three groups; Below Target Profit (<0.8), Within Target Profit (0.8–1.2), and Above Target Profit (>1.2). Similarly, the number of patients seen per month was divided into three tertiles to facilitate categorical analysis. The table showing the three tertiles of patient volume has been included in the Supplementary Materials (Table S1). Chi-square tests revealed statistically significant associations for both suburb density (χ²(6)=106.75, p<0.001) and patient volume (χ²(4)=83.89, p<0.001).

The results reveal a clear pattern: high-density suburban and high-volume practices are overwhelmingly represented in the 'Above Target' profit category, whereas low-density and low-volume practices are predominantly in the 'Below Target' category. We have included these findings as Tables 8 and 9 in the manuscript, which detail the Chi-square tests for suburb density and patient volume, respectively. These findings provide a clear, data-driven explanation for the observed profit differences. The updated de-identified dataset, which includes the new variables (Profit Ratio, Profit Category, and Patient Volume tertiles), has been attached for your reference. We have also included this important finding in the Abstract (Page 1), Discussion (Page 20) and Conclusion (Page 22) sections of our manuscript.

Conclusion

We are grateful for the opportunity to revise our work. We believe these changes improve the rigor and clarity of our manuscript. The submission includes:

• A revised manuscript with tracked changes.

• A clean version of the revised manuscript.

• This point-by-point rebuttal letter.

• Supplementary Materials document with updated questionnaire.

• Updated de-identified dataset.

Please don’t hesitate to let us know if additional clarification is required.

---

## [Decision Letter · Decision Letter 2]

26 Nov 2025

Developing a Pricing Model for General Medical Consultation Services among Private Consulting Rooms in Harare, Zimbabwe

PONE-D-25-17786R2

Dear Dr. Gota,

We’re pleased to inform you that your manuscript has been judged scientifically suitable for publication and will be formally accepted for publication once it meets all outstanding technical requirements.

Kind regards,

Fatima Suleman, PhD

Academic Editor

PLOS ONE

Additional Editor Comments (optional):

Reviewers' comments:

Reviewer's Responses to Questions

**Comments to the Author**

Reviewer #1: All comments have been addressed

2. Is the manuscript technically sound, and do the data support the conclusions?

Reviewer #1: Yes

3. Has the statistical analysis been performed appropriately and rigorously?

Reviewer #1: Yes

4. Have the authors made all data underlying the findings in their manuscript fully available?

Reviewer #1: No

5. Is the manuscript presented in an intelligible fashion and written in standard English?

Reviewer #1: Yes

Reviewer #1: Thank you for addressing my queries in the previous revisions - I am now satisfied with the manuscript and I have no further criticisms or suggestions

**Do you want your identity to be public for this peer review?** For information about this choice, including consent withdrawal, please see our Privacy Policy

Reviewer #1: No

---

## [Editor Report · Acceptance letter]

PONE-D-25-17786R2

PLOS ONE

Dear Dr. Gota,

I'm pleased to inform you that your manuscript has been deemed suitable for publication in PLOS ONE. Congratulations! Your manuscript is now being handed over to our production team.

Kind regards,

on behalf of

Professor Fatima Suleman

Academic Editor

PLOS ONE